# Stereo Video Quality Metric Based on Multi-Dimensional Analysis

**DOI:** 10.3390/e23091129

**Published:** 2021-08-30

**Authors:** Zhouyan He, Haiyong Xu, Ting Luo, Yi Liu, Yang Song

**Affiliations:** 1College of Science and Technology, Ningbo University, Ningbo 315211, China; hezhouyan@nbu.edu.cn (Z.H.); luoting@nbu.edu.cn (T.L.); liuyi@nbu.edu.cn (Y.L.); 2School of Mathematics and Statistics, Ningbo University, Ningbo 315211, China; xuhaiyong@nbu.edu.cn

**Keywords:** stereo video, quality metric, stereo video decomposition, image entropy

## Abstract

Stereo video has been widely applied in various video systems in recent years. Therefore, objective stereo video quality metric (SVQM) is highly necessary for improving the watching experience. However, due to the high dimensional data in stereo video, existing metrics have some defects in accuracy and robustness. Based on the characteristics of stereo video, this paper considers the coexistence and interaction of multi-dimensional information in stereo video and proposes an SVQM based on multi-dimensional analysis (MDA-SVQM). Specifically, a temporal-view joint decomposition (TVJD) model is established by analyzing and comparing correlation in different dimensions and adaptively decomposes stereo group of frames (sGoF) into different subbands. Then, according to the generation mechanism and physical meaning of each subband, histogram-based and LOID-based features are extracted for high and low frequency subband, respectively, and sGoF quality is obtained by regression. Finally, the weight of each sGoF is calculated by spatial-temporal energy weighting (STEW) model, and final stereo video quality is obtained by weighted summation of all sGoF qualities. Experiments on two stereo video databases demonstrate that TVJD and STEW adopted in MDA-SVQM are convincible, and the overall performance of MDA-SVQM is better than several existing SVQMs.

## 1. Introduction

Stereo video can provide a more immersive watching experience for viewers, so stereo video systems have been widely used in various fields [1,2]. The processing in stereo video system will inevitably introduce distortions in stereo videos, which will seriously affect the user experience [3]. Therefore, the research on quality metric is of great significance for performance optimization of stereo video system and has aroused more and more attention [4]. Generally speaking, stereo video quality evaluation can be divided into subjective evaluation and objective evaluation. Among them, subjective evaluation is implemented by evaluate the quality of stereo video according to the watching experience of viewers. Because human eyes are the terminal receivers of stereo video, subjective evaluation is considered to be best in reliability and accuracy. However, such an evaluation process is time-consuming and laborious. Furthermore, subjective evaluation cannot be integrated in practical video systems. Therefore, the objective metric which can evaluate stereo video quality automatically is highly demanded [5].

Considering the dependence of stereo video quality metric (SVQM) on reference video, objective SVQM can be divided into three categories, that is full-reference (FR), reduced-reference (RR) and no-reference (NR) [6]. As the names indicate, FR and NR metrics need to obtain all or part of the undistorted reference video, while NR metric only needs to process the distorted video. Obviously, in a video system, the receiver cannot get information from a reference video. Therefore, NR metric has better practical value. However, due to the lack of reference, the design of NR metric is much more difficult. Compared with FR and RR metrics, its development speed is slow, but has more room for improvement. Hence, this work will attempt to propose an objective NR metric for stereo video quality evaluation.

Benefiting from the rapid development of traditional image and video quality metric, objective quality metrics for stereo video are gradually proposed. In the early stage of the research, the design of SVQM is mainly realized by referring to image quality metric (IQM) and 2D video quality metric (VQM). For example, the well-performed IQM and VQM [7,8,9], such as SSIM, VSI, VQM, etc., are directly implemented to calculate the quality score of each frame or each view, and then the overall quality is obtained by weighted average of the frame or view qualities. Although, there is a certain correlation between the frame quality and the stereo video quality. However, this category of method only focuses on the spatial information in stereo video and ignores the vital role of temporal and inter-view information. Therefore, the performance is obviously not satisfactory. To solve this problem, a series of metrics start focusing on the processing of multi-dimensional information in stereo video. From this starting point, multi-dimensional transformations [10,11], such as 3D-DCT, 3D spatial-temporal structure and 3D structure tensor, have been integrated in SVQM designing. Naturally, such a strategy provides inspiration for the further development of SVQM. Motivated by this strategy, this work will attempt to design an NR-SVQM from the perspective of multi-dimensional information analysis.

In stereo video, the information of each dimension interweaves. Separately processing information in different dimensions cannot reflect the characteristics of stereo video precisely. In this work, to deeply explore the characteristics of stereo video, we attempt to implement multi-dimensional joint analysis, and propose an SVQM based on multi-dimensional analysis (MDA-SVQM). Specifically, firstly, by comparing the correlation in each dimension, the optimal decomposition direction is selected to construct the temporal-view joint decomposition (TVJD) model for stereo group of frames (sGoF). Then, based on decomposition, the feature extraction of TVJD subband information is completed and the quality of sGoF is obtained by regression. Finally, the quality weight of each sGoF is obtained by calculating the spatial-temporal joint energy using spatial temporal energy weighting (STEW) function, and the final stereo video quality can be acquired by weighted summation of sGoF qualities. Experiments on two stereo video databases demonstrated that the MDA-SVQM proposed in this work can accurately evaluate the quality of stereo video.

To sum up, the innovation of this work mainly lies in:(1)To solve the problem that stereo video contains both inter-view and temporal information, which leads to the difficulty in quality evaluation, this work designs a TVJD model based on correlation analysis, which can describe the characteristics of stereo video more accurately;(2)To solve the problem that TVJD subbands are substantial and complex, this work classifies the subbands by their own characteristics and generation mechanism. On this basis, the features of different subbands are extracted separately, which makes the features more sensitive to distortions;(3)To solve the problem that it is difficult to model the influence of temporal fluctuation on stereo video quality, in this work, STEW function is designed by simulating the stimulation of spatial-temporal alteration on the visual system.

The rest of this paper is organized as follows. Section 2 reviews the current research status of SVQM. The proposed MDA-SVQM is introduced in detail in Section 3. In Section 4, the performance of MDA-SVQM is tested and compared with the existing metrics in stereo video database, and experimental results are further analyzed and discussed. Finally, Section 5 summarizes this work and points out the future research direction.

## 2. Related Works

In this section, we will make a brief review of SVQM and related fields.

As the terminal receiver of visual information, viewer’s subjective rating is obviously of better accuracy. Because of this, subjective quality evaluation plays an important role in exploring the characteristics of stereo video. In order to ensure the standardization and reasonability of subjective evaluation, video quality expert group (VQEG) in International Telecommunication Union (ITU) has released a series of standards for subjective evaluation implementation, such as ITU-R BT.500, ITU-R P.910 and ITU-R BT.1788. In these standards, subjective evaluation methods such as ACR-HR, DSCQS and SAMVIQ and watching environment requirements are recommended in detail. In the early stage of research, researchers mainly analyzed the impact of different acquisition and display devices on stereo visual quality. In these researches, Saad et al. [12] studied the effects of resolution and display mode on the subjective quality of stereo video. Ijsselstein et al. [13] studied the influence of different camera parameter settings on stereo video quality. At the same time, explorations on the relationship between equipment performance and subjective quality haveare also been carried out by subjective evaluation [14]. However, with the continuous development of stereo video processing, the subjective evaluation is no longer limited on the impact of hardware devices on stereo video quality, and subjective evaluations begin to pay more attention on the relationship between video content and video quality. Seuntiens et al. [15] studied the quality influence of different compression ratio and spatial resolution of two view. In addition to understanding the characteristics of stereo video, another important role of subjective evaluation is to establish a subjective quality database. Due to the limitation of acquisition and display devices, the study of stereo video database is far behind the traditional quality database. Among the few publicly available stereo video databases, Uroy et al. [16] established NAMA3D-COSPAD 1 stereo video database, which provides distorted videos with H.264 encoding compression, JPEG2000 (JP2K) encoding compression, image sharpening, resolution reduction and down sampling. In establishing of the database, the researchers concluded that the distortion such as down sampling and image sharpening will not significantly affect the quality of stereo video, which sheds light on objective metric designing.

Based on the databases established by subjective evaluation, the objective metrics for stereo video haves been proposed gradually. As mentioned in Introduction, VQM is the foundation of designing SVQM. Starting from simply combining the spatial quality of each frame in a video, state-of-the-art VQMs employ deep learning and other signal processing methods [17], which show outstanding quality evaluation performance. However, the ignorance of stereo information aroused the attention of researchers. Gradually, with the exploration of visual psychology, SVQMs start to focus on simulation of visual perception, especially by modelling the binocular perception of disparity information in human visual system (HVS). Specifically, Battisti et al. [18] proposed an HVS perception-based SVQM. According to depth information and binocular rivalry, they generated cyclopean frames for both reference and distorted stereo videos, then traditional video quality metric was performed to yield the quality score. Galkandage et al. [19] calculated the energy scores to measure the spatial quality by perception model, and then pooled these spatial qualities by empirical methods to obtain the overall quality of a stereo video. Yu et al. [20] proposed an SVQM considering temporal characteristics of video by motion intensity and binocular perception in HVS. In [21], a novel SVQM was proposed by modelling visual attention and just-noticeable difference for distortions. In addition, with the machine learning and neural network widely applied in image and video processing, SVQMs based on machine learning had achieved significant development. Jiang et al. [22] utilized random forest to establish the mapping from motion feature in tensor domain to stereo video quality. In [23], Narwaria et al. extracted structural information features by singularly valuable decomposition, and then proposed a Support Vector Regression (SVR)-based metric. Furthermore, Yang et al. [24] introduced a 3D convolutional neural networks-based framework for stereo video quality evaluation. To sum up, the above metrics integrated machine learning techniques in the traditional feature extraction and perception model. This strategy not only ensures the consistency of quality metric and visual perception, but also effectively improves the performance of SVQMs.

## 3. Proposed MDA-SVQM

Figure 1 illustrates the framework of MDA-SVQM, which consists of four modules: sGoF decomposition, Subband feature extraction, sGoF quality prediction and stereo video quality pooling. In sGoF decomposition, by interleaved analyzing temporal and inter-view correlation of each sGoF, TVJD is adopted to decompose sGoF into different subbands with the optimal decomposition structure. Then, according to the characteristics of each TVJD subband, statistical-based and the local organization of image direction (LOID)-based feature extraction are implemented to construct the feature vector. Furthermore, an SVR-based sGoF quality prediction model is applied to obtain sGoF qualities. Finally, the STEW function is designed for sGoF quality pooling, and stereo video quality can be acquired. In this section, we will describe each module of MDA-SVQM in detail.

To improve the readability, the summary of some important notations and abbreviations in this paper are given in Table 1 and Table 2.

### 3.1. TVJD for sGoF Decomposition

Essentially, stereo video can be characterized as high-dimensional (usually four-dimension). To be specific, one dimension along view direction, one dimension along temporal direction, and two dimensions in spatial domain. Obviously, different dimensional information will cause different visual perception in HVS. Meanwhile, different frequency components will also percept differently. As a result, each frequency component decomposed from different dimensions should be processed separately to complete the precise description of stereo video. Based on above analysis, considering the computational complexity, in this subsection, temporal and inter-view lifting-based wavelet transform has been jointly implemented as sGoF decomposition.

In standard lifting-based wavelet transform, the pixels in the same location of two adjacent frames are decomposed into high and low frequency components. However, when viewing a video, the viewer will track the moving objects. Hence, to make the decomposition more consistent with visual perception, compensation is integrated, so as to realize the object-based frequency decomposition.

Let *I_l_*_1_ and *I_l_*_2_ denote two adjacent frames in a video, and *M_l_*_1__→l2_ be the mapping for two frames. Then, the object-based decomposition can be implemented to decompose *I_l_*_1_ and *I_l_*_2_ into high frequency subband *H_l_*_1,*l*2_ and low frequency subband *L_l_*_1,*l*2_, which can be expressed as
(1)$$\{\begin{array}{l} H_{l1,l2}[x,y]=I_{l2}(x,y)-\{M_{l1\to l2}(I_{m})[x,y]+M_{l1\to l2}(I_{l2})[x,y]\}/2\\ L_{l1,l2}[x,y]=I_{l1}(x,y)+\{M_{l1\to l2}(H_{l1})[x,y]+M_{l1\to l2}(H_{l1})[x,y]\}/4 \end{array},$$
where (*x*, *y*) represents the pixel location, and *H_m_* and *L_m_* are the high and low frequency component decomposed from *I_l_*_1_ and *I_l_*_2_.

From the perspective of construction, stereo video can be regarded as the ordered arrangement of frames in both temporal and inter-view directions. Resultingly, the frequency decomposition of stereo video should also be implemented in two directions, which are defined as temporal filter (TF) and view filter (VF). Meanwhile, as Equation (1) illustrated, the key problem in decomposition is how to establish the mapping relationship between adjacent frames. Considering the characteristics of stereo video, motion vector can be used as the mapping in temporal direction, and disparity vector can be adopted for inter-view direction. Hence, the compensation can be optimized by motion compensation (MC) and disparity compensation (DC). To sum up, MCTF and DCVF can be used to decompose stereo video in temporal and inter-view directions.

Obviously, a stereo video contains two views, so only one inter-view decomposition is needed. As for temporal decomposition, the number is determined by the temporal length of the sGoF (usually 16 frames, i.e., four times that of decompositions). Hence, for efficiency and accuracy, it is necessary to determine the decomposition order between two directions reasonably. Generally speaking, in multi-view video, temporal correlation is usually stronger than inter-view correlation. Therefore, a conventional decomposition structure is to implement DCVF after multi-level MCTF. Specifically, let *V* represent inter-view decomposition and *T* represent temporal decomposition. For an sGoF with 16 frames, the traditional structure can be expressed as 4*T* + *V*. However, due to the variety of scene depth and motion intensity, the conclusion that inter-view correlation is stronger is not always true. Meanwhile, in multi-level decomposition, the correlation changes after each decomposition. That is, the decomposition in one direction can reduce the correlation in that direction to 1/2, without affecting the correlation of the other directions. Therefore, to decompose sGoF more efficiently, it is highly necessary to design an adaptive interleaving decomposition structure for temporal and inter-view decomposition. Aiming at this, a correlation analysis-based TVJD model is designed in this work.

In detail implementation, the key problem is how to determine the optimal direction in each time of decomposition. According to the calculation of object-based decomposition given in Equation (1), it can be deduced that, for ideal mapping of two adjacent frames *I_m_* and *I_m_*_+1_, the amplitudes of high frequency subband coefficients are close to 0. That is to say, the smaller the amplitude of high frequency subband coefficients, the stronger the correlation between two frames to be decomposed. Furthermore, because the decomposition scheme adopted in this work integrated motion and disparity compensation, the amplitude of motion vector and disparity vector can also be utilized to reflect the correlation in temporal and inter-view directions. According to the above two conclusions, a decomposition cost function can be constructed from two aspects to determine the correlation of each direction, that is the amplitude of high frequency subband and motion or disparity vectors, which are denoted by *H* and **V***_d_*, respectively. Let *D_cost_* represent the decomposition cost, the calculation can be expressed as
(2)$$D_{cost}(d)=\frac{1}{m\times n}[\sum_{x=1}^{m}\sum_{y=1}^{n}H(x,y)+\omega \cdot \sum_{x=1}^{m}\sum_{y=1}^{n}V_{d}(x,y)],$$
where *d* denotes the decomposition direction, that is *T* or *V*. (*x*, *y*) denotes the pixel location and *m* and *n* is the spatial height and width of the stereo video.

Then, the decomposition costs of both directions can be calculated before each level of decomposition, respectively, to adaptively select the optimal direction. Let *S_optimal_* denote the optimal direction, the selection can be expressed as
(3)$$S_{optimal}=\arg\min D_{cost}(d),$$

It should be pointed out that only two views are included in stereo video. Consequently, inter-view decomposition only needs to be implemented once. Therefore, for low-computational complexity in TVJD, once the inter-view decomposition is selected as the optimal, the decomposition cost will not be calculated in the subsequent decomposition, and the temporal decomposition will be selected directly.

To conclude, by the calculation and comparison of decomposition costs in different directions, TVJD constructs the adaptive interleaving decomposition structure for sGoF, thus improving the decomposition efficiency and describing the multi-dimensional characteristics of stereo video more accurately.

### 3.2. TVJD Subband Feature Extraction

Referring to scalable image processing, in TVJD, it is only necessary to further decompose the low frequency subband. Accordingly, in MDA-SVQM, a 16-frame sGoF can be decomposed into six subbands by TVJD. After that, further exploration for subband characteristics is needed, so that feature extraction can be more consistent with visual perception.

As a basis, several stereo videos with different scenes are selected for analyzing. Because TVJD adopts joint decomposition in temporal and inter-view directions, we employ temporal index (TI) and disparity index (DI) to represent the characteristics of the test video. To be specific, the larger the TI is, the greater the temporal difference is, that is, the motion in the scene is more intensity. Meanwhile, the larger the DI is, the greater the difference in the inter-view direction is, that is, the stereo of the scene is more.

In Table 3, the relationship between video characteristics and TVJD subband is analyzed by using TI and DI. Obviously, it can be seen that with the decrease in motion intensity, represented by the decrease in TI, the intensity of subbands containing temporal high frequency information decreases, while the intensity of low frequency subbands increases. Similarly, with the decrease in stereo sense, represented by the decrease in DI, the intensity of subbands containing view high frequency information decreases, while the intensity of subbands containing view low frequency information increases. According to the above analysis, the TVJD subbands can be divided into four categories, namely MFJI, MRJI, SFJI and SRJI. In the next, we will extract the corresponding features according to subbands characteristics and visual perception to reflect the impact of distortion on them.

#### 3.2.1. High Frequency Subband Feature Extraction

In sGoF, the high frequency subbands generated by TVJD, that is MFJI and MRJI, usually represent the detail information of moving objects. Meanwhile, for natural scene image, the distribution of high frequency components usually follows a pattern of high peak and heavy tail. Based on the above analysis, by analyzing the relationship between distribution and distortion, features of the MFJI and MRJI can be extracted.

Figure 2 shows the distribution of several high frequency subbands from stereo videos encoded with different QPs. Specifically, (a), (b) and (c) represent high frequency subbands generated from encoded video with QP of 32, 38 and 44. Obviously, with the increase in QP, the coding distortion will be aggravated, which will lead to the quality degradation. Meanwhile, for a certain encoded video, TVJD generated high frequency subbands in different levels, which are denoted by different colors. By observing Figure 2, firstly, all high frequency subbands follow a similar distribution, which means that the same feature extraction can be adopted for all high frequency subband. Then, with the gradual increase in video compression, the distribution of high frequency subband changes regularly, that is, peak raised and tail suppressed. For analysis, due to the increase in compression, detail information in the video scene will be lost, resulting in more 0 values of the coefficients in the motion information subband, which is expressed as the peak rise in histogram. Consequently, to reflect the distortion degree, we can quantify the loss of scene information by calculating its histogram kurtosis. Let *K_MF_* (*m*) and *K_MR_* (*m*) denote the kurtosis feature of MFJI and MRJI in *m*-th sGoF. Detail calculation can be expressed as
(4)$$\{\begin{array}{l} K_{MF}(m)=\frac{\sum_{i=1}^{N}\left(\varphi_{MF}(m)-\bar{\varphi_{MF}}\right)/N}{\delta_{MF}{\left(m\right)}^{4}}\\ K_{MR}(m)=\frac{\sum_{i=1}^{N}\left(\varphi_{MR}(m)-\bar{\varphi_{MR}}\right)/N}{\delta_{MR}{\left(m\right)}^{4}} \end{array},$$
where *φ_MF_*(*m*) and *φ_MF_*(*m*) are coefficients of MFJI and MRJI, respectively, $\bar{\varphi_{MF}}$ and $\bar{\varphi_{MR}}$ denote the coefficients mean value of the two subbands.

On the other hand, the compression can also reduce the richness of texture information in video, which leads to the amplitude diversity decrease in motion information coefficients. In histogram, it can be reflected by the reduction in the tail. Meanwhile, the greater the amplitude diversity of the subband coefficients, the greater the overall information entropy. Therefore, in MDA-SVQM, information entropy is applied to represent the richness of motion information subbands. Let *E_MF_* and *E_MR_* denote the information entropy of subbands MFJI and MRJI. Detail calculation can be expressed as
(5)$$\{\begin{array}{l} E_{MF}=-\sum_{i=1}^{N}p[\varphi_{MF}(i)]\log p[\varphi_{MF}(i)]\\ E_{MR}=-\sum_{i=1}^{N}p[\varphi_{MR}(i)]\log p[\varphi_{MR}(i)] \end{array},$$
where *p*[*φ_MF_*(*i*)] is the probability of coefficients *φ_MF_*(*i*).

To sum up, by exploring the relationship between coefficient distribution and distortion degree, the kurtosis and entropy of each motion information subband are extracted as the distortion features. Then, the distortion feature vectors of motion information subband of are constructed. Let **V**_M_ denotes the distortion feature vector of motion information subband, the construction can be expressed as
**V**_M_ = [*K*_*MF*,1_, *K*_*MR*,1_, *E*_*MF*,1_, *E*_*MR*,__1_, *K*_*MF*,1_, *K*_*MR*,1_, *E*_*MF*,1_, *E*_*MR*,1_],(6)
where the subscripts 1 and 2 indicate the two high frequency subbands generated by TVJD.

#### 3.2.2. Low Frequency Subband Feature Extraction

Different from high frequency subband, low frequency subband mainly represents structure information in stereo video scene. To further prove the high correlation between low frequency subband and video scene, Figure 3 analyzes the correlation from both subjective and objective aspects. Subjectively, the low frequency information basically contains most of the scene information in the original video. On the other hand, from the perspective of information entropy, the correlation between the low frequency subband and the original scene can also be seen quantitatively.

In order to more accurately measure the distortion, feature extraction should be consistent with the visual perception for low frequency information. As the key of visual understanding, LOID features can joint-capture the position and direction information of the object in the scene, and have the advantages of rotation invariance. Therefore, the low dimensional representation of low frequency subband based on LOID features is obviously a method that conforms to HVS perception for low frequency information.

In this work, a steerable wavelet machine (SWM) [25] is adopted to obtain the LOID feature for low frequency subband. In general, the input frame is convolved with a family of Circular harmonic wavelets (CHWs), and then processed by two levels of non-linear *steermax* operations to obtain the LOID feature. Let *f_i_* denote the *i*-th frame in low frequency subband, the feature map of *f_i_* can be acquired through SWM layers, which is denoted as *t_i_*. For detail implementation, *f_i_* is first convolved with CHWs, *ϕ*^(*n*)^. Then, the convolution coefficients are mapped to an initial gradient-based moving frame (MF) representation, and processed by the first level of non-linear *steermax* operation, which can be expressed as
(7)θ(1)=argmaxθ∈[0,2π)(Re(〈F,ϕs,0(1)(R−θ(−x))〉)),
where $R_{-\theta }$ is used for rotation, which can be expressed as $\left[\begin{array}{cc} \cos(\theta ) & -\sin(\theta )\\ \sin(\theta ) & \cos(\theta ) \end{array}\right]$. Meanwhile, *s* denotes the wavelet scale, and *F* is defined as local geometry.

Then, the combination of CHWs is learned and the class-wise templates can be constructed, which is denoted as *ψ*
^(*c*)^. Meanwhile, the final level of *steermax* operation can be implemented as
(8)$$\theta_{}^{\left(2\right)}=\underset{\theta \in [0,2\pi )}{\arg\max}\left(\langle F,\psi_{s,0}^{\left(c\right)}(R_{-\theta }(-x))\rangle \right),$$

Eventually, the final feature representation *t_i_* can be constructed by the combination of the results of the above-mentioned two levels of *steermax* operation as
(9)$$t_{i}=\left[steer_{s,x}(f_{i}{}^{},\theta_{}^{\left(1\right)},\theta_{}^{\left(2\right)})\right],$$
where *steer* ( ) represents the *steermax* operation.

However, the LOID in Equation (9) is still a low-dimensional representation of low frequency subband. To precisely reflect the influence of distortion on stereo video, it is necessary to further explore the characteristics of LOID. From the above implementation of LOID feature extraction, it can be found that MF-based LOID performs a lot of gradient-based calculation. Therefore, low frequency subband representation can thus be considered as gradient-based. Since the gradient contrast plays an important role for image recognition in HVS, when the distortion occurred, it will inevitably affect the gradient contrast. Meanwhile, the image gradient histogram follows the Weibull distribution, which also has a high correlation with the visual perception. To sum up, in this part, Weibull distribution-based statistical feature is extracted for low frequency subband.

Previous studies on visual physiology have pointed out that the shape and scale parameters in Weibull distribution’s probability density function (PDF) can accurately describe the spatial consistency and complexity of images [26]. Hence, in this work, the shape and scale parameters of LOID representation are employed to reflect the distortion of low frequency subband. The Weibull distribution PDF of LOID can be expressed as
(10)$$p(GM_{LOID};\beta ,\gamma )=\frac{\beta }{\eta }{\left(\frac{GM_{LOID}}{\eta }\right)}^{\beta -1}\exp(-{\left(\frac{GM_{LOID}}{\eta }\right)}^{\beta }),$$
where *η* and *β* denote the scale and shape parameters respectively, and *GM_LOID_* is the gradient amplitude of the LOID feature map. Let *p_x_* and *p_y_* denote the Prewitt filter operators in the directions of horizon and vertical, the calculation of *GM_LOID_* can be expressed as
(11)$$GM_{LOID}=\sqrt{{\left(t_{i}*p_{x}\right)}^{2}+{\left(t_{i}*p_{y}\right)}^{2}},$$
where * is used to convolve LOID feature with Prewitt filter operators in two directions.

Specifically, the scale parameter *η* represents the width of the distribution and can reflect the local contrast. The change of local contrast will affect the perception of human eyes on quality. At the same time, the shape parameter *β* refers to the kurtosis of the distribution, which is sensitive to the local edge. In order to further explain the relationship between the parameters and distortion, we encoded the same image with JPEG and JP2K in different compression degrees, and extracted the corresponding parameters of each compressed image. Figure 4a,b illustrate the tendency of parameters with different distortion degrees. In detail, Dis1 to Dis6 in horizontal axis represent six compressed images with different distortion degrees, and the vertical axes denote the value of scale or shape parameter of each distorted image, respectively. As can be seen from the Figure 4, with the increase in distortion degree, the edge weakens and the contrast decreases. As a result, the parameters gradually decrease. It can also be concluded from the results that the parameters of the same image change regularly with the increase in distortion. In conclusion, the scale and shape parameters of Weibull distribution can be used to describe and measure distortion.

Based on the above conclusion, for the low frequency subband, firstly, the LOID feature map reflecting its essential information is extracted by SWM. Then, considering the perceptual characteristics of Weibull distribution, the PDF parameters of the LOID feature map are taken as distortion features. Let **V***_s_* represent the distortion feature vector, the construction of the vector can be expressed as
(12)$$V_{s}=[\eta_{SF,t},\eta_{SR,t},\beta_{SF,t},\beta_{SR,t}],$$
where *η**_SF_*, *η_SR_*, *β_SF_* and *β**_SR_* are the scale and shape parameters of each low frequency subband, respectively.

#### 3.3. sGoF Quality Prediction

By obtaining the distortion features of each TVJD subband, the final feature vectors can be constructed, which is denoted by $\vec{V_{sGoF}}$, $\vec{V_{sGoF}}=\left[V_{M},V_{s}\right]$. Then, sGoF quality prediction can be implemented by establishing the mapping between the feature vector and subjective quality of sGoF. In this subsection, SVR is employed to train for the sGoF quality prediction model, denoted as *P_sGoF_* ( ). In SVR training, a radial basis function is adopted. Specifically, we used a publicly available LIBSVM package for SVR implementation. Then, *P_sGoF_* ( ) can be used to predict the quality of sGoF *q_sG_*. The calculation can be expressed as
(13)$$q_{sG}=P_{sGoF}(\vec{V_{sGoF}}),$$

It should be noted that a traditional subjective quality database only provides the overall subjective quality of each distorted stereo video, but not the subjective quality of each sGoF. Therefore, to obtain the sGoF subjective quality, we build a new stereo video subjective quality database, and consider the overall quality of each stereo video as the sGoF’s subjective quality. Meanwhile, we make two assumptions to guarantee the rationality of the above scheme. Firstly, because of its short temporal duration, the sGoF with small length can be considered as stable in temporal quality. Secondly, because the quality of each sGoF in our database changes periodically, and the period is set as the same length of sGoF in this work. To conclude, it can be considered that the overall subjective quality can be used to represent the sGoF’s subjective quality.

### 3.4. sGoF Quality Pooling

In Section 3.3, the quality of each sGoF is obtained by sGoF quality prediction model. Then, sGoF quality pooling should be implemented to get the final stereo video quality. To solve this problem, purposely-designed STEW is adopted in MDA-SVQM.

Early studies on binocular visual psychology pointed out that for simple ideal stimuli, with the increase in contrast on a certain view, the dominance of the view will gradually increase [27]. For watching stereo videos, the stimuli fluctuation not only occurs between left and right views, but also in temporal domain. Furthermore, the enhancement of contrast represents that the intensity of the stimuli is increasing, and image energy can be used to reflect the signal intensity. Hence, this work generalizes the above conclusion in temporal domain and proposes an STEW function. Specifically, in STEW, the video can be considered as a two-dimensional arrangement of single stimuli, while the stimuli in spatial domain can be regarded as the weight of the temporal stimuli. That is, the spatial weight is used as the weighting basis of the temporal weight, so the spatial and temporal weights are combined to form the joint weights.

Next, to calculate the STEW of each frame, we need to obtain the spatial and temporal weights of each pixel. In detail implementation, firstly, the local energy of pixel in location (*i*, *j*) is calculated by Gaussian weight function, which can be expressed as
(14)$$e_{d}(i,j)={\left[\omega_{ij}{\left(x(i,j)-\mu \right)}^{2}\right]}^{1/2},$$
where the subscript *d* represents the direction, and it can be expressed as the temporal energy and spatial energy, which are denoted by *e_T_* and *e_S_*, respectively. *μ* is local average value, *μ* = *∑ ω_ij_ x*_(*i*, *j*)_. *ω_ij_* is the local energy weight calculated by Gaussian weighting function. Specifically, the spatial weight is calculated by 11 × 11 two-dimensional Gaussian weight, while the temporal weight is calculated by 1 × 11 weight function.

By obtaining the temporal and spatial local energy of each pixel, the temporal and spatial weight of a pixel in the location of (*i*, *j*) of frame *n* can be calculated as
(15)$$\{\begin{array}{c} W_{T}(n)=e_{T}{\left(n\right)}^{2}/(\sum_{n=1}^{N}e_{T}{\left(n\right)}^{2})\\ W_{S}(i,j)=e_{S}{\left(i,j\right)}^{2}/(\sum_{i,j}^{}e_{S}{\left(i,j\right)}^{2}) \end{array},$$

Since the STEW is used to weight video quality in temporal domain, the spatial weight is employed as the basis of temporal weighting. Thus, spatial weight and temporal weight are combined to form the spatial-temporal joint weight. Therefore, the spatial-temporal weights can be calculated. Let *W_S−T_* denote the STEW weights, it can be calculated as
(16)$$W_{S-T}=\frac{\sum_{i=1,j=1}^{H,W}W_{S}(i,j)\cdot {\left[W_{T}(i,j)\right]}^{\alpha }}{\sum_{i=1,j=1}^{H,W}W_{S}(i,j)},$$

In MDA-SVQM, the temporal information within an sGoF have been considered in TVJD. Here, we only need to consider the temporal information between sGoFs. As analyzed previously, most of the scene information in stereo video is contained in low frequency subbands. Therefore, when calculating the quality weight of each sGoF, the low frequency subbands of each sGoF can be reorganized as a three-dimensional information, and the weight of each frame can be calculated by STEW.

Finally, the weight of each frame in the three-dimensional information is considered as the weight of each sGoF. The final stereo video quality can be obtained by STEW weighting of the sGoF quality. Let *Q_sV_* denote stereo video quality, it can be expressed as
(17)$$Q_{sV}=\frac{\sum_{i=1}^{N}W_{S-T}(i)\cdot q_{sG}(i)}{\sum_{i=1}^{N}W_{S-T}(i)},$$

## 4. Experimental Results and Analysis

In this section, we will first introduce the stereo video database and performance indicators used in this work for performance evaluation. Then, the performance of each module in MDA-SVQM is verified to prove the effectiveness of the proposed framework. Finally, the overall performance of MDA-SVQM will be further illustrated by the performance comparison with other existing SVQMs. It should be noted that the quality metrics are realized in MATLAB^®^ R2014a, and a computer with Intel(R) Core (TM) i7-3770 CPU @3.40 GHz, 8G RAM, Windows 7 64-bit is used for the verification of metrics in two databases.

### 4.1. Stereo Video Database and Perforamnce Indicators

In our work, a stereo video database is set up by subjective experiment, which is named as NBU-3DV database. Here, the NBU-3DV database is utilized to train for sGoF quality prediction model. Then, the validation experiment and performance comparison are carried out in the internationally acknowledged NAMA3D-COSPAD1 database (abbreviated as NAMA3D database) [16]. At the same time, a series of performance indicators are employed to evaluate the performance of objective metrics. Here, we will briefly introduce the corresponding database construction and performance indicators.

#### 4.1.1. NAMA3D-COSPAD1 Stereo Video Database

The NAMA3D database contains 10 undistorted stereo videos with a resolution of 1080 × 1920 at 25 fps as reference. Meanwhile, the database also contains 100 symmetric distorted stereo videos generated from reference videos. Specifically, five types of distortion are considered in the NAMA3D database, including H.264 compression, JP2K compression, reduction in resolution, image sharpening and down-sampling with sharpening. The mean opinion score (MOS) with a range of 1 to 5 is used to represent the subjective quality of distorted video. The higher the MOS value is, the better the quality of distorted video is.

#### 4.1.2. NBU-3DV Stereo Video Database

As previously mentioned, in this work, an sGoF quality prediction model should be constructed by SVR. Generally speaking, it is not reliable to share the same dataset for training and prediction in regression. Based on the above considerations, we establish an NBU-3DV database by subjective experiment. In this section, we will briefly introduce the subjective experiment and NBU-3DV database construction.

In construction of the database, firstly, we select six pairs of undistorted stereo video as reference videos, which are all publicly available standard stereo video for encoding test. The reference videos have HD or full HD resolution with a YUV 420 format in .avi containers of a 10-s time duration. In Figure 5, first frames in left view of reference stereo videos in NBU-3DV database are illustrated. In order to comprehensively and accurately test SVQMs’ performance for all types of videos, we fully considered the intensity of video motion and the complexity of scene information in sequence selection. In Figure 6, the SI and TI scatter plots of all reference videos in NBU-3DV are given, which are calculated by averaging the SI and TI of left and right view, respectively. Obviously, the points’ distribution in Figure 6 is scattered, which proves the rationality of reference video selection in NBU-3DV database.

As for distorted video generation, we use HEVC [28,29] to encode the reference stereo videos. This distorted stereo video generation method is mainly based on two considerations. Firstly, with the increase in resolution, the efficient compression of stereo video is highly demanded. Meanwhile, HEVC is the most commonly used encoding standard for stereo video. In detail implementation, by changing the reference direction (intra coding/random access) and QP (QP = 26/32/38/44), a total number of 48 pairs of stereo videos with different distortion degrees can be obtained (6 reference videos × 2 reference direction × 4 QPs = 48 distorted videos). In the subjective experiment, 20 subjects were selected to watch the distorted stereo video and scored. Specifically, 20 subjects were asked to watch the test stereo videos. For stereo video display, Samsung 3DTV (model number: UA65F9000) was chosen for stereo display. In detail, we played the test video in 3D mode of the TV, and the viewer watched through electronic 3D glasses with a TV adapter. Meanwhile, the Absolute Category Rating with Hidden Reference (ACR-HR) on five discrete scores has been performed as the subjective scoring method. Similar with NAMA3D database, a MOS value with a range of 1 to 5 is used to represent the quality of distorted stereo video. For better illustration, the distorted stereo video in NBU-3DV database, Figure 7 provides the frames in distorted stereo videos encoded with different QPs and corresponding MOS scores obtained from subjective evaluation. Meanwhile, we have enlarged the region where the balloon is located in the image, so that distortion degree of different videos can be easily seen.

#### 4.1.3. Performance Indicators

Three commonly used measures are applied to quantitatively evaluate the performance of the quality metric, that is Pearson linear correlation coefficient (PLCC), Spearman rank correlation coefficient (SROCC) and Root mean squared error (RMSE). The illustration and detail calculation of the performance indicators are as follows.

(a)PLCC

PLCC reflects the linearity between subjective quality and quality evaluated by an objective quality metric. The value range of PLCC is −1 to 1. The closer the absolute value of PLCC is to 1, the better the evaluation performance of the objective metric. Otherwise, the worse. The calculation of PLCC can be expressed as
(18)$$PLCC=\frac{\sum_{i=1}^{n}\left(oq_{i}-\bar{oq})(sq_{i}-\bar{sq}\right)}{\sqrt{\sum_{i=1}^{n}(oq_{i}-\bar{oq})\sum_{i=1}^{n}\left(sq_{i}-\bar{sq}\right)}},$$
where *n* represents the total number of distorted stereo video. *oq_i_* and *sq_i_* denote the objective and subjective quality of *i*-th distorted video, and $\bar{oq}$ and $\bar{sq}$ are the average objectived and subjective quality of all distorted video.

(b)SROCC

SROCC is employed to measure the monocity between subjective quality and objective quality. The value range of SROCC is [−1, 1]. When subjective and objective qualities are strictly monotonic, the absolute value of SROCC is 1. The SROCC can be calculated as
(19)$$SROCC=1-\frac{6\cdot \sum_{i=1}^{n}{\left(Roq_{i}-Rsq_{i}\right)}^{2}}{n\cdot (n^{2}-1)},$$
where *Roq_i_* and *Soq_i_* denote the rank of *i*-th objective and subjective quality in all objective and subjective qualities, respectively.

(c)RMSE

RMSE quantitatively reflect the numerical distance between subjective and objective quality. RMSE values from 0 to infinity. Obviously, the smaller the RMSE, the better the performance of objective metric. Detail calculation can be expressed as
(20)$$RMSE=\sqrt{\frac{1}{n}\sum_{i=1}^{n}{\left(oq_{i}-sq_{i}\right)}^{2}},$$

In summary, for a perfect objective quality metric, PLCC and SROCC should be close to 1, and RMSE should approximate to 0.

### 4.2. Verification of Each Module in MDA-SVQM

MDA-SVQM mainly consists of four modules, including stereo video decomposition, feature extraction, GoF quality prediction and stereo video quality pooling. Among them, the innovation of this work mainly lies in TVJD, as well as the new STEW weighting function. Therefore, the effectiveness of TVJD and STEW is verified in this subsection.

#### 4.2.1. Verification of TVJD

In MDA-SVQM, TVJD is used for sGoF decomposition. To achieve the purpose of efficient decomposition, TVJD adaptively selects the optimal decomposition direction according to the scene characteristics. To prove the scene adaptability of TVJD, we choose stereo video Boxer from NAMA3D database for testing in this part. Figure 8 shows the order of inter-view decomposition in optimal decomposition structure selected by TVJD for each sGoF in video Boxer.

As previous mentioned, temporal correlation is stronger than intra-view correlation for most stereo video. Hence, it can be seen from Figure 8 that intra-view decomposition is usually performed after temporal decomposition in some of the earlier sGoFs. However, from sGoF 14 to sGoF 19, the priority of intra-view decomposition is continuously improved. In order to better illustrate and analyze such changes, the first frames in the left view of sGoF 1 and sGoF 14 are shown in Figure 8. It can be found that only one person moves in sGoF 1, and the motion intensity is not severe. Hence, the temporal correlation is stronger than intra-view correlation. Consequently, the structure of 4*T* + *V* is selected by TVJD, that is, the inter-view decomposition is implemented after all temporal decomposition. However, from sGoF 14, a second person appears, and the moving intensity has been greatly increased compared with that before. Therefore, from this sGoF, the temporal correlation gradually decreases, and the intra-view correlation gradually exceeds the temporal correlation. As for the optimal decomposition structure, the order of inter-view decomposition gradually advanced, and the structure of 2*T* + *V* + 2*T* or 3*T* + *V* + *T* are chosen. Through the above experimental results and analysis, it can be concluded that TVJD can adaptively construct the optimal decomposition structure according to correlation in two directions and lay a good foundation for feature extraction and quality evaluation.

#### 4.2.2. Verification of STEW

In MDA-SVQM, STEW is employed to weight the sGoF quality and yield the final stereo video quality. Obviously, since the inter-view-temporal information in stereo video has been processed in TVJD, STEW only considers the spatial-temporal information in sGoF weighting. Based on the above premise, in this part, we adopt LIVE video quality database for validation of STEW. To reflect the performance of STEW in temporal weighting more accurately, we design several different evaluation schemes for comparison. In detail implementation, the SSIM is applied to calculate the image quality of each frame in a video. Then, different weighting strategies are adopted for temporal quality pooling. To be specific, temporal average pooling, temporal asymmetric pooling [18] and temporal fluctuation pooling [30] are chosen. Finally, for quantitative comparison, we tested all the temporal pooling strategies on the LIVE video database. The performance indicators are listed in Table 4. The experimental results demonstrate that when the same metric is used for each frame’s quality evaluation, the temporal weighting of STEW can significantly affect the performance of the quality metric. Obviously, the experimental results also show that compared with other temporal pooling strategies, the proposed STEW is more consistent with the human eye’s perception of spatial-temporal information, to improve the evaluation performance of video quality metric.

### 4.3. Overall Performance Evaluation

In this subsection, we evaluate the performance of the proposed MDA-SVQM in NAMA3D database and NBU-3DV databases, and performance comparison with other existing well-performed SVQM is also conducted. In MDA-SVQM, the sGoF quality prediction model is established by SVR. In order to ensure the reliability of the performance evaluation, we repeat the train-test procedure for 100 times and select the median value of performance indicators as the final performance indicators. At the same time, two different methods are used in training datasets selection:(1)Randomly select the distorted stereo videos in NAMA3D database to construct the test dataset, that is, 80% of the distorted video is used as the training data, and the remaining 20% is the training dataset, and no overlap between two sets;(2)Adopt NBU-3DV database as the training dataset and test on NAMA3D database.

By the above implementation, it can be ensured that the indicators obtained in the performance verification experiment are accurate and reliable to fully reflect the performance of the quality metric. In comparative methods selection, we first select the traditional IQM (PSNR and SSIM), and then a series of existing well-performed SVQM is also contained. With regards to 2D metrics, we apply them on each view and frame, then take the average value as the approximate quality of stereo video.

Firstly, Table 5 shows the performance comparison between the MDA-SVQM and existing quality metric in NBU-3DV database. In addition, the metric noted with cross-database represents that the performance indicators are obtained by cross database training. Theoretically, for the listed metrics, the performance of PSNR should be worse than SSIM, and MNSVQM should report the highest indicators. Obviously, the results in NBU-3DV database are consistent with the theoretical analysis, which proves that the construction of the database is reliable. Consequently, the sGoF quality prediction model trained by the database can accurately evaluate the quality of each sGoF. Meanwhile, MDA-SVQM jointly considers multi-dimensional information. Hence, compared with the traditional metrics, MDA-SVQM has a significant improvement in evaluation performance. In all, the experimental results in this part not only prove the accuracy and reliability of the NBU-3DV database, but also verify the performance of proposed MDA-SVQM.

On the other hand, Table 6 shows the performance comparison of MDA-SVQM and a series of existing well-performed metrics on the internationally acknowledged NAMA3D database. For better visualization, the two indicators with the best performance are marked in bold. It should be noted that MDA-SVQM-1 represents that sGoF quality prediction model uses the distorted stereo video in NAMA3D database for both training and testing. While MDA-SVQM-2 uses the distorted stereo videos in NBU-3DVdatabase as the training dataset and uses the distorted videos in NAMA3D database as the test dataset. It can be seen from this table that the proposed method performs better than other methods when cross database strategy is not used for training and test. Of course, to improve the reliability of performance evaluation, even if the cross-dataset method is used for training and test, the performance indicators of proposed method is better than most SVQA methods. That is, it can still maintain high evaluation performance. The comparison of the above experiments proves that proposed method has good stability and universality.

Meanwhile, in order to further prove the extensibility of MDA-SVQM, the performance indicators for different types of distorted video in NAMA3D database are given in Table 7, and the boxplots of different performance indicators are also provided in Figure 9. It should be pointed out that there are only 10 pairs of distorted stereo videos in the distortion type of reduction in resolution, sharpening, down-sampling and sharpening. To ensure the stability of the test results, we combine these three types of distortion and those denoted as D&S distortions. The experimental results demonstrate that MDA-SVQA is well-performed on different type of distortion in NAMA3D database, which confirms that MDA-SVQA works well for different types of distortion.

### 4.4. Discussion

In the above experiments and analyses, we verify the accuracy and validity of the proposed MDA-SVQM. To prove the positive role of each module in MDA-SVQM, we designed different experimental schemes. First, by analyzing the decomposition direction of TVJD for a specific test video, we prove that TVJD can adaptively select the decomposition direction according to the scene characteristics, to achieve the purpose of efficient decomposition. On the other hand, for sGoF quality weighting, we prove the validity of STEW by comparing with the traditional temporal pooling strategies on video quality. Finally, by comparing the performance with the existing well-performed SVQM on NAMA3D and NBU-3DV databases, it is shown that the MDA-SVQM has excellent stereo video quality evaluation performance.

Based on this work, there is still plenty room for further analysis and research. Firstly, in decomposition for sGoF, although the purposely designed TVJD decomposes the corresponding pixels in inter-view or temporal direction by matching and compensation, which improves the decomposition efficiency. However, the correlation between pixels and pixels in the spatial domain is ignored. That is, decomposition is not implemented for objects in the scene. Hence, in subsequent research, we can attempt to integrate semantically related decomposition strategies, which can achieve object-based decomposition, thereby further enhancing the performance of decomposition. Meanwhile, in MDA-SVQM, for the subbands generated by TVJD, the corresponding feature extraction strategies are selected by analyzing the composition and generation mechanism of the subbands. Although the proposed method considers the validity of subband information and feature extraction to some extent, there is room for further improvement. On the one hand, it is not comprehensive to analyze the nature of subband coefficients only from the perspective of generation mechanism. Obviously, even in the subbands of the same frequency component, there will be differences in the properties because of the different content they contained. Therefore, further studies will attempt to do a more detailed and comprehensive subband analysis from the perspective of the scene objects and the motion in the video. On the other hand, MDA-SVQM extracts features from the perspective of statistical characteristics. Although SWT features can be used to reflect the intrinsic characteristics of coefficients, they are essentially a gradient-based feature extraction strategy. Therefore, in the future research, we can attempt to analyze the high-dimensional information in the TVJD subband from the perspective of deep learning and data mining, so that feature extraction can more accurately reflect the distortion of stereo video.

## 5. Conclusions

Stereo video contains both stereo information generated by left and right views and spatial-temporal information generated by video, which makes it difficult to describe in the design of quality metric. Aiming at this problem, in this work, we propose a new quality metric based on multi-dimensional analysis (MDA-SVQM). Specifically, by analyzing temporal and inter-view correlation, the optimal decomposition structure is adaptively constructed to complete the decomposition of sGoF by TVJD. Then, according to the characteristics of different subbands, the subbands are classified and the corresponding quality features are extracted. At the same time, sGoF quality is obtained by regression. Finally, an STEW function is designed to weight the sGoF quality, so as to obtain the stereo video quality. Experimental results on NAMA3D database and NBU-3DV databases show that MDA-SVQM is reasonable and effective in structure, better in performance than the existing SVQM, and can accurately measure the distortion of stereo video. In the future work, TVJD can be further analyzed and improved. Firstly, an object-based decomposition can be attempt to design for better consistency with visual perception. Secondly, the characteristics of TVJD subbands should also be further explored, so as to extract the corresponding subband feature more accurately. Another, future study will focus on integrating MDA-SVQM in stereo video coding, especially for coding parameter optimization. Thus, the proposed metric can eventually contribute to the watching experience improvement.

## Figures and Tables

**Figure 1 entropy-23-01129-f001:**
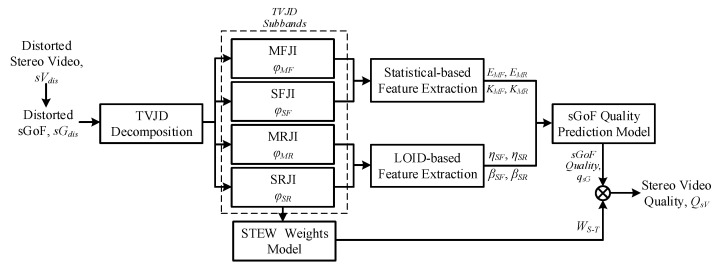
Framework of proposed MDA-SVQM.

**Figure 2 entropy-23-01129-f002:**
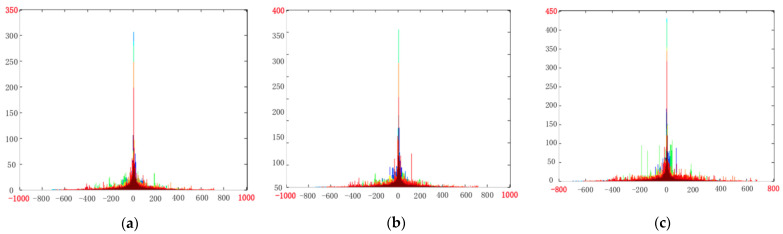
Distribution of high frequency subband generated by TVJD of encoded stereo video with different QPs by different QPs: (**a**) QP = 32; (**b**) QP = 38; (**c**) QP = 44.

**Figure 3 entropy-23-01129-f003:**
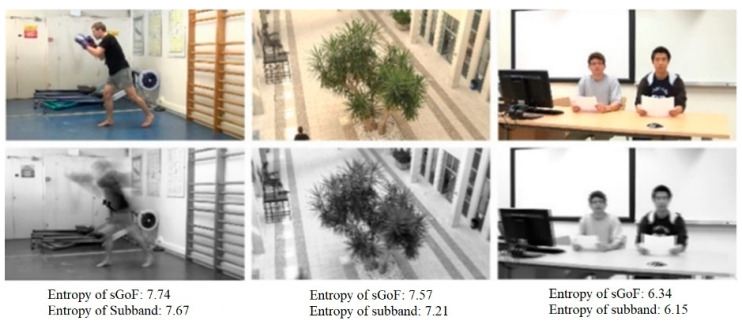
Entropy Correlation between low frequency subband information and original video scene.

**Figure 4 entropy-23-01129-f004:**
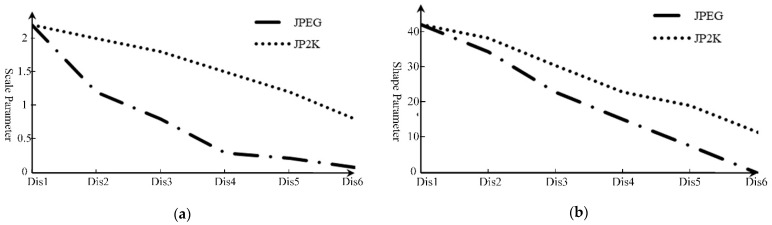
Tendency of Weibull’s PDF parameters with different encoding distortion degree: (**a**) Tendency of scale parameter; (**b**) Tendency of shape parameter.

**Figure 5 entropy-23-01129-f005:**
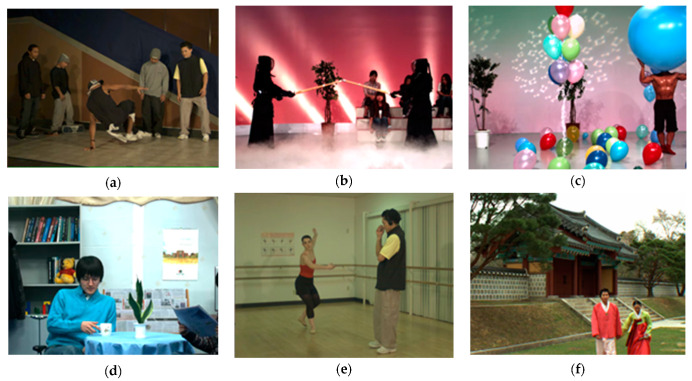
First frames in left view of reference stereo videos in NBU-3DV database: (**a**) Break dancer; (**b**) Kendo; (**c**) Balloons; (**d**) Newspaper; (**e**) Ballet; (**f**) Lovebird1.

**Figure 6 entropy-23-01129-f006:**
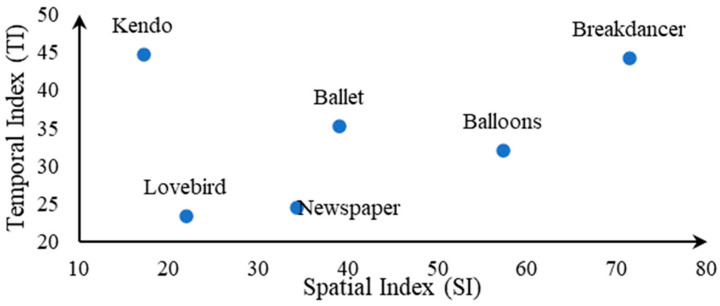
Scatter plots of SI and TI for reference video in NBU-3DVdatabase.

**Figure 7 entropy-23-01129-f007:**
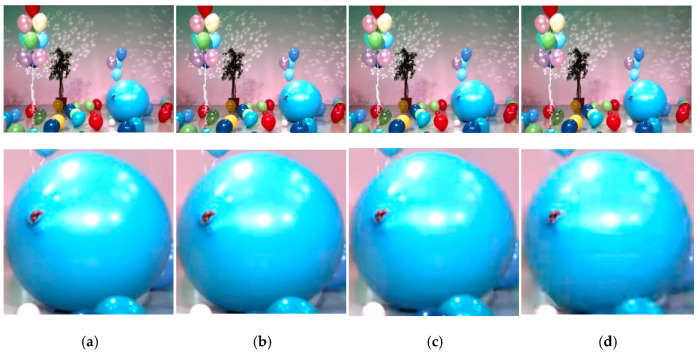
Distorted stereo video frames in NBU-3DV database with corresponding subjective qualities: (**a**) QP = 26, MOS = 4.64; (**b**) QP = 32, MOS = 3.71; (**c**) QP = 38, MOS = 2.92; (**d**) QP = 44, MOS = 2.05.

**Figure 8 entropy-23-01129-f008:**
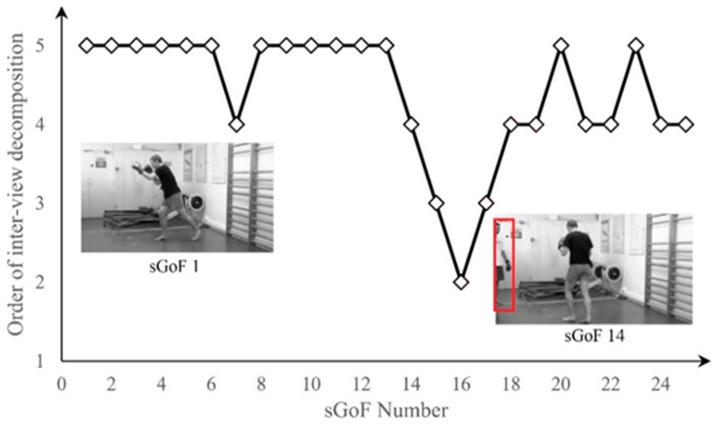
The change of inter-view decomposition in each sGoF of stereo video Boxers.

**Figure 9 entropy-23-01129-f009:**
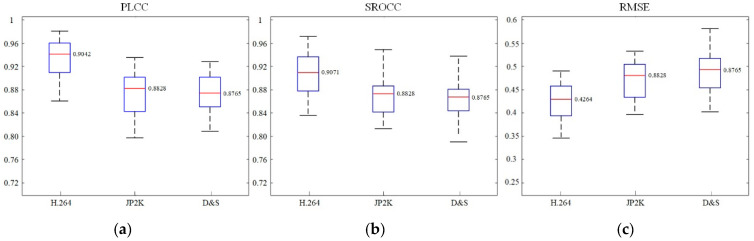
Boxplots of performance indicators of MDA-SVQM in NAMA3D database: (**a**) PLCC; (**b**) SROCC; (**c**) RMSE.

**Table 1 entropy-23-01129-t001:** Important abbreviations and corresponding full names.

Abbrev.	Full Name	Roles in MDA-SVQM
TVJD	Temporal-view Joint Decomposition	sGoF decomposition
SFJI	Static-Fusion Joint Information	Subbands generated by TVJD
MFJI	Motion-Fusion Joint Information	
SRJI	Static-Rivalry Joint Information	
MRJI	Motion-Rivalry Joint Information	
SWM	Steerable Wavelet Machine	LOID extraction algorithm
LOID	Local organization of image direction	Feature map of SFJI and SRJI
STEW	Spatial-temporal Energy Weighting	sGoF weight calculation

**Table 2 entropy-23-01129-t002:** Important notations and definitions.

Notation	Definition
$SV_{dis}=[V_{L,dis},V_{R,dis}]$	Distorted video, consists of *V_L_*_,*dis*_ and *V_R,dis_*
$sG_{dis}(i)=[G_{L,dis}(i),G_{R,dis}(i)]$	*i*-th distorted sGoF, consists of *G_L_*_,*dis*_(*i*) and *G_R_*_,*dis*_(*i*).
$\varphi_{MF},\varphi_{SF},\varphi_{MR},\varphi_{SR}$	Subbands of MFJI, SFJI, MRJI and SRJI
$K_{MF},K_{MR}$	Kurtosis of MFJI and MRJI distribution, used as features of MFJI and MRJI, feature length: 1 × 1
$E_{MF},E_{MR}$	Entropy of MFJI and MRJI coefficients, used as feature of MFJI and MRJI, feature length: 1 × 1
$\Theta_{\phi ,\psi }$	*steermax* operation denoted with angle maps used in SWM
$t_{i,SF},t_{i,SR}$	LOID Feature maps of SFJI and SRJI generated by SWM
$\eta_{SF,t},\eta_{SR,t}$	Scale parameter of Weibull distribution used as feature of SFJI and SRJI, feature length: 1 × 1
$\beta_{SF,t},\beta_{SR,t}$	Shape parameter of Weibull distribution used as feature of SFJI and SRJI, feature length: 1 × 1
$V_{S}$	Feature vector of high frequency subband, vector length: 1 × 4
$\vec{V_{sGoF}}=\left[V_{M},V_{S}\right]$	Feature vector of an sGoF, vector length: 1 × 12
$q_{sG}(i)$	Quality of *i*-th sGoF *sG*(*i*)
$w_{S}(i,j)$	Spatial energy weights of pixels in location (*i*, *j*)
$w_{T}(i)$	Temporal energy weight of *i*-th sGoF
$W_{S-T}(i)$	Spatial-temporal joint energy weights of *i*-th sGoF
$Q_{sV}$	Quality of distorted stereo video

**Table 3 entropy-23-01129-t003:** Relationship between TVJD subband and scene characteristics of several test video. All test videos are selected from NAMA3D database.

Test Video	TI	DI	Inter-View Low Frequency	Inter-View High Frequency	Temporal Low Frequency	Temporal High Frequency
Boxers [16]	129.28	9.66	182.36	33.25	155.63	20.68
Hall [16]	73.64	12.86	165.65	40.26	170.22	10.65
News Report [16]	54.57	8.06	188.39	28.48	180.65	5.65

**Table 4 entropy-23-01129-t004:** Performance indicators of quality metrics with different temporal pooling strategies in LIVE video database.

Temporal Pooling Strategy	PLCC	SROCC	RMSE
Average Pooling	0.7065	0.6947	0.7769
Asymmetric Pooling [20]	0.7181	0.7077	0.7698
Fluctuation Pooling [30]	0.7115	0.7054	0.7709
STEW based Pooling	0.7216	0.7092	0.7510

**Table 5 entropy-23-01129-t005:** Performance Indicators of Different Metrics on NBU-3DV Database.

Metric	PLCC	SROCC	RMSE
PSNR	0.7663	0.7419	0.7254
SSIM [7]	0.8006	0.7934	0.5217
MNSVQM [22]	0.8998	0.8846	0.4608
MDA-SVQM	0.9328	0.9226	0.3562
MDA-SVQM (cross-database)	0.9004	0.8892	0.4627

**Table 6 entropy-23-01129-t006:** Performance Indicators of Different Metrics on NAMA3D Database.

Metrics	PLCC	SROCC	RMSE
PSNR	0.6699	0.6470	0.8433
SSIM [7]	0.7664	0.7492	0.7296
VQM [9]	0.6340	0.6006	0.8784
PHVS-3D [10]	0.5480	0.5146	0.9501
3D-STS [11]	0.6417	0.6214	0.9067
BSVQE [19]	0.8124	0.8009	0.4952
Metric in [21]	0.6503	0.6229	0.8629
MNSVQM [22]	0.8545	0.8349	0.4538
MDA-SVQM-1	0.8884	0.8765	0.4451
MDA-SVQM-2	0.8765	0.8698	0.4489

**Table 7 entropy-23-01129-t007:** Performance indicators of different types of distortion in NAMA3D database.

Distortion	PLCC	SROCC	RMSE
H.264	0.9042	0.9071	0.4264
JP2K	0.8828	0.8796	0.4773
D&S	0.8765	0.8711	0.4902

## Data Availability

Data is contained within the article.

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
