# Peer review of "Stereo Video Quality Metric Based on Multi-Dimensional Analysis"

_entropy, 2021, doi:10.3390/e23091129_

Round 1

Reviewer 1 Report

The paper has a small similarity index (<17%) and some novelty in solving:

-The problem of the stereo video contains both inter-view and temporal information, which leads to difficulty in quality evaluation, this work designs a TVJD model based on correlation analysis, which can describe the characteristics of stereo video more accurately. The problem of TVJD subbands is substantial and complex, this work classifies the subbands by their own characteristics and generation mechanism. On this basis, features of different subbands are extracted separately, which makes the features more sensitive to distortions. The difficult problem to model the influence of temporal fluctuation on stereo video quality, in this work, STEW function is designed by simulating the stimulation of spatial-temporal alteration on the visual system. Some comments to improve the paper:

-Figure 1 the framework is not clear, a high-resolution figure must be provided.

-Figure 3, what represents the different colors?

-Many English languages are incorrect. 

-The abstract must be revised because not support clearly the body of the paper.

-The introduction should be more clear that present the objective of the study, methodology, and results.

-There is a large number of grammatical errors and tenses must be unified.

The issue of clarity in the writing is serious as it leads the reader too confusing at every step in the way, trying to understand what is really meant and how it connects to the paragraph and the previous ideas.

-line 523, the authors took the median, why not the mean?

Author Response

Thanks for your comments.

Reviewer 2 Report

I think the authors have submitted an interesting manuscript which has potential, although the reviewer has some comments and concerns.

1.) In general, the English of the paper is satisfactory but please check the English of the manuscript once again because there are minor solecisms in the text. For example: "stereo video systems has been widely used in various fields" -> Stereo video systems have been widely used ...

2.) I think the introduction and related work sections are slightly poorly structured. It would be nice to read something about how the videos are evaluated subjectively in practice. In the introduction section in the first paragraph, the authors just remark that subjective evaluation is "time-consuming, laborious, and poor practicability". Why? What are the major standards for subjective video quality assessment?

3.) The related works section is not bad but it would be nice to read something about general objective video quality assessment at least in one paragraph. Major trends and approaches, etc. The following papers could be cited in this paragraph: a) Götz-Hahn et al.: KonVid-150k: A dataset for no-reference video quality assessment of videos in the wild, IEEE Access, 2021 b) D. Varga: Multi-pooled Inception features for no-reference video quality assessment, VISAPP, 2020

3.) In Section 3, the authors propose a lot of novel features. To help the readers, the proposed features and the main characteristics of the features (input, length, etc.) should be summarized in a Table before the detailed description of the features.

4.) Unfortunately, Eq. 1. is obscure for the reviewer. The authors should explain the meaning of variables H, L, M, I, etc. or draw a figure if the authors find a figure more effective than formal definitions.

5.) In Eq. 2., variable H is not explained.

6.) Figure 5 depicts two graphs. However, the numerical values are missing from both axises.

7.) I think the authors present a decent amount of experimental results in Section 4. I think the authors should depict the MOS distributions in applied databases because it would be interesting for the readers.

8.) In Section 4.1.3, the formulas of PLCC and SROCC should be given. Moreover, it would be nice to explain why two different correlation coefficients are required to characterize the performance of the VQA methods.

9.) A box plot about the measured PLCC, SROCC, and RMSE values would be interesting in the experimental results section (https://www.mathworks.com/help/stats/boxplot.html).

10.) Implementation details are missing from the manuscipt (applied programming languages, libraries, hardware configuration, etc.).

Author Response

Thanks for your comments

Reviewer 3 Report

The paper presents SVQ-metric for multi-dimensial analysis (MDA). The paper is enhanced both in theoretic and experimental sides. The article is of high quality and will be very interesting to the readers.

Some points for improvement:

Fig,2 contain both Images and a table. 

Figs 4, 6 and 7 can be enlarged.

Conclusions. Future work can be extended. Moreover, the authors can provide applications of their work.

Reference/Bibliography: The authors can update the references and include some new. Also, please include relative articles from Electronics Journal of MDPI.

Author Response

Thanks for your comment.

Reviewer 4 Report

Stereo video contains both stereo information generated by the left and right views and spatiotemporal information generated by the video, which makes it difficult to describe accurately in the design of a quality metric. To solve this problem, the authors propose in this work a new quality metric based on a multidimensional analysis.

Nevertheless, they should not overshadow recent work related to H265/HEVC [1,2] which should be analyzed and cited.

[1] Ferroukhi, M.; Ouahabi, A.; Attari, M.; Habchi, Y.; Taleb-Ahmed, A. Medical Video Coding Based on 2nd-Generation Wavelets: Performance Evaluation. Electronics 20198, 88.

[2] B. Mallik, A. Sheikh-Akbari and A. Kor, "HEVC Based Mixed-Resolution Stereo Video Codec," in IEEE Access, vol. 6, pp. 52691-52702, 2018, doi: 10.1109/ACCESS.2018.2870183.

Moreover, relations (7), (8), (9), (10) are not sufficiently justified, and relation (11) is badly written, the * sign should be replaced by x.

Figure (5) is well known and is therefore not original

Author Response

Thanks for your comment.

Round 2

Reviewer 2 Report

The authors modified the manuscript according to the reviewer's comments. The authors answered my questions. I propose this manuscript for publication.

Reviewer 4 Report

The manuscript has been considerably improved. It deserves to be published as is.